# Determination of Optimal Predictors and Sampling Frequency to Develop Nutrient Soft Sensors Using Random Forest

**DOI:** 10.3390/s23136057

**Published:** 2023-06-30

**Authors:** Muhammad Arhab, Jingshui Huang

**Affiliations:** Chair of Hydrology and River Basin Management, Technical University of Munich, Arcisstrasse 21, 80333 Munich, Germany; m.arhab@tum.de

**Keywords:** surrogate, water quality, nitrate, machine learning, optimization

## Abstract

Despite advancements in sensor technology, monitoring nutrients in situ and in real-time is still challenging and expensive. Soft sensors, based on data-driven models, offer an alternative to direct nutrient measurements. However, the high demand for data required for their development poses logistical issues with data handling. To address this, the study aimed to determine the optimal subset of predictors and the sampling frequency for developing nutrient soft sensors using random forest. The study used water quality data at 15-min intervals from 2 automatic stations on the Main River, Germany, and included dissolved oxygen, temperature, conductivity, pH, streamflow, and cyclical time features as predictors. The optimal subset of predictors was identified using forward subset selection, and the models fitted with the optimal predictors produced R^2^ values above 0.95 for nitrate, orthophosphate, and ammonium for both stations. The study then trained the models on 40 sampling frequencies, ranging from monthly to 15-min intervals. The results showed that as the sampling frequency increased, the model’s performance, measured by RMSE, improved. The optimal balance between sampling frequency and model performance was identified using a knee-point determination algorithm. The optimal sampling frequency for nitrate was 3.6 and 2.8 h for the 2 stations, respectively. For orthophosphate, it was 2.4 and 1.8 h. For ammonium, it was 2.2 h for 1 station. The study highlights the utility of surrogate models for monitoring nutrient levels and demonstrates that nutrient soft sensors can function with fewer predictors at lower frequencies without significantly decreasing performance.

## 1. Introduction

Monitoring water quality is vital for the proper management of valuable freshwater resources for drinking water supply, recreational purposes, and ecosystem support. With the increasing impact of climate change and human activities on water quality, it has become increasingly important to establish monitoring schemes that can provide continuous real-time information on water quality and track its event-scale changes, seasonal patterns, and long-term variations. Among others, nutrients are one of the main focuses of water quality monitoring because of their relevance to unresolved eutrophication problems and further environmental consequences.

The technological advancement in the field of sensors science has allowed for in situ measurements without manual sampling [1,2]. Ion-selective electrodes (ISE), wet-chemical analyzers, and optical sensors are the three categories in which current commercially available sensor technologies for nutrient analysis fall [3]. ISE are cheap and simple to use, however they have been criticized for being imprecise and prone to disruptions and straying. Wet chemical analyzers and optical sensors, on the other hand, exhibit superior resolution and accuracy, but they are costly and have high maintenance needs [4]. Their utility for long-term in situ and online outdoor monitoring may be constrained by these shortcomings [5].

The above problems can be addressed using soft sensor technology based on machine learning (ML) models. The soft sensor is a feasible and economical alternative to expensive or impractical physical measurement sensors [6,7]. In recent years, the development of soft computing tools based on ML models in the fields of water and environmental engineering has received wide attention [8,9,10,11,12]. In this regard, decision-tree-based algorithms such as Random Forest (RF) are widely used due to their high generalization ability. Francke et al. (2008) is one of the earlier studies that employed the RF to predict suspended sediment concentrations [13]. They compared it with a generalized linear regression model and found the RF to be more robust and it performed well to reproduce sediment dynamics. Ha et al. (2020) using RF found R^2^ higher than 0.8 for orthophosphate, nitrite and nitrate in Vietnam (Ha, Nguyen [14]). Shen et al. (2020) used the RF to predict total nitrogen and total phosphorous covering 62,495 stations across the United States and obtained R^2^ between 0.56 and 0.88 (Shen, Amatulli [15]). Harrison et al. (2021) used the RF models to predict different nitrogen and phosphorus species; their models of total nitrogen and total phosphorous were able to explain variations of 85% and 74%, respectively (Harrison, Lucius [16]). Among all these studies, the study of Tran et al. (2022) sets the premise of this study, where they used 15-min high-frequency measurements of nutrients and other water quality parameters and developed the soft sensor using the random forest algorithm and found the R^2^ higher than 0.95 for NO_3_-N, OPO_4_-P and NH_4_-N [17]. These studies demonstrate that soft sensors based on ML models can provide reliable alternative real-time monitoring. However, in order to build an effective and efficient soft sensor, the ML models require the appropriate amount of input data of high quality, while balancing cost and feasibility [18].

To minimize overfitting and hence improve generalization, it is usually recommended to pick a subset of predictors in any ML model [19]. Furthermore, a simpler model with fewer predictors is preferable since it will take less time and resources to train and test and will be easier for the end-users to understand [20]. When building soft sensors, using fewer predictors also means lower monitoring costs for practical reasons. The subset specifies the combination of surrogate physical sensors that will be used in the future to operate the soft sensors. The goal is to use the fewest number of predictors possible to save money on the expense of procuring, installing, and maintaining the surrogate physical sensors on which the soft sensors work. However, the selection for the smallest optimal predictor subset in the process of building soft sensors has not yet become a standard procedure. Furthermore, surrogates that are part of the optimal subset for several target variables can be shared across models. The surrogate that is used a greater number of times would naturally become more important. This would help in determining the surrogate sensors that are needed at the stations, as well as provide experience in establishing soft sensors even across stations in large monitoring networks.

Higher sampling frequency is highly useful as it can be used for a variety of applications such as baseline characterization, event-based monitoring, compliance monitoring, and forecasting models for water quality [21,22,23,24]. However, high data volumes are not always advantageous, as the capacity to manage, store, and process the data must also be taken into account, along with any related expenses [25]. When data are gathered too infrequently, they will depict an erroneous or incomplete pattern of water quality, but when they are collected too frequently, they will be redundant, expensive, and contain noise [26]. The frequency of data must be balanced against their manageability, although the exact balance will depend on the overall goal of any monitoring scheme [27]. Additionally, these measurement frequencies require previous setup and have little leeway when it comes to reacting to unplanned events.

Precisely quantifying the sampling frequency requirements of physical sensors required for developing soft sensors would yield significant benefits in terms of reducing operational costs and conserving energy while simultaneously improving the quality of data fed into the models. Researchers have used a variety of methods for the determination of optimal sampling frequency (OSF), and they all yielded different results depending on the use of monitoring systems. Zhou (1996) determined that 1 month was the OSF for a ground-water monitoring system for a pumping station using Nyquist frequency [28]. According to the macro-index level and Mann–Kendall test used by Naddeo et al. (2007) to compare the ecological state of the rivers, OSF was determined to be 1 month [29]. Anvari et al. (2009) used a qualitative approach to find that the OSF of measuring stations set up to provide data for river water quality modeling systems was 15 min [30]. For river water quality monitoring systems, Liu et al. (2014) suggested a grab sampling every 2 to 3 months [27]. Khalil et al. (2014) evaluated the level of ambient water quality based on a historical time series of 36 years measured from 23 different locations [31]. They suggested varying sampling frequencies for several variables, ranging from 4 to 12 samples each year. Chen and Han (2018) used a novel qualitative technique on a surface water quality monitoring system and reported 5 min as the OSF [32]. Silva et al. (2019) used spectral analysis and suggested varying sampling frequencies based on the size of the watershed [33]. However, to our knowledge there is no study that has determined the OSF required for the soft sensors to train on.

The aim of this study is to develop a robust and efficient soft sensor for the surrogating high-frequency nutrient concentrations using other physical sensor measurements. This study attempts to establish the effective and efficient soft sensors through two aspects: (1) identifying the optimal subset of predictors to eliminate redundant predictors and (2) determining the OSF of the sensors required for model training to obtain the best performance to operational cost ratio. Reducing the number of sensors and running them at lower frequency would reduce operational costs. This study facilitates the development of leaner soft sensors with less redundancies and lower costs and promotes real-life applications of the soft sensors.

## 2. Materials and Methods

In this study, random forest models were developed to predict the nutrients. The models were optimized for the number of predictors and then trained on different sampling frequencies of the physical sensors to determine the OSF of data the models need for their optimal performances. To construct the models, data from two automated measuring stations on the Main River, namely Erlabrunn (upstream) and Kahl am Main (downstream), were used. Four water quality parameters—dissolved oxygen (DO), temperature (Temp), electronic conductivity (EC) and pH—along with two time features—month and week—and discharge rate (Q) at each station were used as predictors. The model targets were the concentrations of nitrate (NO_3_-N), orthophosphate (OPO_4_-P), and ammonium (NH_4_-N). The workflow of this study is presented in Appendix A.

### 2.1. Data and Study Site Description

The data used in this study were obtained from the Waterscience service of Bavaria (Gewässerkundlicher Dienst Bayern, https://www.gkd.bayern.de/ (accessed on 7 May 2023)) of the Bavarian State Office for the Environment (Bayerisches Landesamt für Umwelt, LfU, Augsburg, Germany). Two water quality monitoring stations in the Bavarian section of the Main River representing upstream (Station Erlabrunn, Erlabrunn, Germany) and downstream (Station Kahl am Main, Karlstein am Main, Germany) were chosen (Appendix A). The Main is the Rhine’s third largest tributary. At its mouth, it discharges water at a mean rate of 225 m^3^ s^−1^. Using automated measuring instruments, Station Erlabrunn and Station Kahl am Main monitor DO, Temp, EC, pH, NO_3_-N and OPO_4_-P with a 15-min temporal resolution, with station Kahl am Main additionally also monitoring NH_4_-N. The discharge rates are not monitored at the same automated water quality stations. Instead, the discharge data from stations that are nearby and are on the mainstream are used, specifically Station Würzburg for Erlabrunn (upstream) and Station Obernau Aschaffenburg for Kahl am Main (downstream).

The surrogate models for the Erlabrunn station were built with a dataset of ~4 years, starting from 15 November 2016 and ending on 30 December 2023. Kahl am Main station does not measure NH_4_-N, so the models were only developed for NO_3_-N and OPO_4_-P. For Kahl am Main, the NO_3_-N model used a dataset of ~4 years between 16 September 2017 and 31 August 2021. The OPO_4_-P and NH_4_-N models used a ~2-year dataset between 2 September 2019 and 31 August 2021.

### 2.2. Tools Used for Data Analysis

The data were in Comma-Separated Values (CSV) format. All data analysis and model development were carried out with Python3.8-based open-source tools and frameworks. Pandas (1.1.3), NumPy (1.19.2), SciPy (1.5.2), and Scikit-learn (0.23.2) are the major libraries utilized in this study. The chosen libraries are used extensively in the community and have well-prepared documentation openly available. The individual functions are explained in the following text.

### 2.3. Data Pre-Processing and Spliting

Data pre-processing is important because the dataset has outliers and gaps in measurement. These arise from the maintenance of sensors or when the calibration goes off. For cleaning the outliers, Isolation Forest was used, an outlier detecting function from Scikit-learn library [34]. An expected percentage of contamination is input into the function and based on that the function cleans out the outliers. For NO_3_-N, the database is assumed to be 10% and for NH_4_-N and OPO_4_-P, 20% are assumed as contaminated.

The dataset contains data gaps in monitoring that are indicated by NaN (Not a Number). NaN values are the consequence of maintenance or calibration operations that are regularly performed at the stations, in addition to being produced by deleting outliers. Because the models can not handle NaN values, they must be eliminated in the dataset before being used in the models. To take advantage of target values as much as possible, the rows with target values as NaN were first eliminated, and the NaN values in the predictors were filled using linear interpolation. A limit of 30 was applied to the number of consecutive NaNs that could be interpolated and if the gap was higher than 30 consecutive NaNs then those readings were removed. To explain further, taking the case of NO_3_-N at Erlabrunn station, NO_3_-N has 3937 NaN values, so all these 3937 rows were removed. Next, the NaN values of the other predictors were interpolated if the number of consecutive NaNs was below 30, or otherwise deleted. This finally created a data frame with 65,808 rows and the statistics that give an overview of the cleaned dataset are in Table 1.

In addition to the four physio-chemical parameters measured by sensors, time is introduced as a feature as well to harness the periodicity of the data. Two time features—month and week—were used. To simulate their cyclic behavior they were transformed using sin and cos functions, similar to the methodology developed by Hempel et al. (2020) [35]. The intrinsic property of sine and cosine adding up to 1 ensures that a circle is formed, and hence the cyclic property of months and weeks is mathematically incorporated into the model. This ensures that the model treats January and December as consecutive months and not as a year apart. This whole exercise gives us four additional predictors, namely week_sin, week_cos, month_sin and month_cos.

The entire dataset was split into working and testing datasets. The latter was made by randomly selecting 20% of the observations from each month. The remaining 80% was the working dataset. The working subset was used to develop the model. Cross-validation was carried out within the working dataset using the Sk-learn. Shuffle split function which used 5 runs, where each run was allocated 20% of the data for validation. The R^2^ and RMSE for the cross-validation performance were calculated by averaging the results of all 5 cross-validation runs. The testing dataset was used to evaluate the final model’s performance on the unseen data.

### 2.4. Random Forest Model

In this study, Random Forest was chosen because of its simplicity and easily explainable mechanism [20]. As the name indicates, a random forest is made up of a huge number of individual decision trees that work together as an ensemble. Each tree in the random forest produces a class prediction, and the most voted class becomes the prediction of our model. The wisdom of crowds is the basic principle of random forest, and itis a simple yet effective one. The reason why the random forest model functions so well is that it consists of a huge number of largely uncorrelated models (trees) that work together to outperform any of the individual constituent models. While some trees may be incorrect, many others will be accurate, allowing the trees to move in the correct direction as a group.

The RandomForestRegressor function of the Scikit-Learn package was utilized in this study, which builds multiple regression trees on different sub-samples of the dataset and employs averaging to increase predicted accuracy and control over-fitting. The models were run on a specified grid of values to determine the optimal value of the hyperparameters. This procedure was carried out using Scikit-Learn’s GridSearchCV function and cross-validation using 5 iterations of ShuffleSplit function on 5 and 50 trees (fewer trees to increase the speed of the searching process). Table 2 lists the hyperparameters and their values found in the grid. These yielded 162 combinations, multiplied by the 5 folds to yield 810 fits. Following the fitting of the hyperparameters, the number of trees was modified by examining the model’s performance with 1, 10, 50, 100, and 200 trees. The greatest number of trees that improved the test error by more than 5% over the results of previous number of trees was picked. Finally, the RF model performances were evaluated using R^2^ and RMSE on the unseen testing dataset and compared with the Multi-Linear Regression (MLR) models with the same predictors.

### 2.5. Selection of Variable Predictors

Various variable selection approaches, such as Best Subset Selection, Hierarchical Partitioning and Stepwise Selection with Bootstrapping have previously been used in water sciences modelling [13]. Stepwise Forward Subset was used in this study, which evaluates the model performance with each individual variable and iteratively more the variable that produces the greatest improvement according to a specific Goodness-of-fit criterion, namely R^2^ from the cross-validation results, in this case ma2.6. for determination of the optimal sampling frequency

All 5 RF models for estimating nutrients were trained on 40 different sampling frequencies, ranging from monthly to 30-min intervals. The dataset for different frequencies were prepared by resampling from the original 15-min-interval dataset. For example, for the hourly dataset, we resampled it such that it was 1 datapoint for every 4 datapoints. The performance of each model with different sampling frequencies was evaluated with the testing dataset, which constituted 20% of the entire dataset with the temporal resolution of 15-min intervals. A graph was plotted using the model evaluation results from the 40 runs, visualising the relationship between the sampling frequency and the Root Mean Squared Error (RMSE). The Kneedle algorithm was then applied to identify the knee point for determining the OSF frequency at which an increase in the frequency no longer resulted in a significant improvement in performance [36]. For the purpose of defining the knee point, Satopaa et al. (2011) in their Kneedle algorithm used the mathematical definition of curvature for a continuous function [36]. There is a common closed-form formula, *K_f_*(*x*), that expresses the curvature of any continuous function f as a function of its first and second derivative at any point, as follows:(1)$$K_{f}\left(x\right)=f^{\prime \prime }\left(x\right)\frac{f^{\prime \prime }\left(x\right)}{{\left(1+f^{\prime }{\left(x\right)}^{2}\right)}^{1.5}}$$

Since curvature is a mathematical measure of how far a function deviates from a straight line, the point of maximum curvature is ideally matched to the ad hoc approaches which operators employ to determine a knee point. As a result, the leveling-off effect that operators employ to detect knees is captured by maximum curvature. The concept of curvature in this application is different than the other work in the area as it is application neutral, does not rely on the link between system characteristics and performance, and does not necessitate the creation of system-specific thresholds.

## 3. Results

### 3.1. Model Performance with Varying Subset of Predictors

The increment in R^2^ was chosen as the determining criteria to select the order in which the predictors were fed into the model. The input variables are described in detail, together with the sequence in which they were chosen, in Table 3. Overall, the performance of the models in terms of R^2^ for both the stations to predict NO_3_-N is higher than the prediction of NH_4_-N and OPO_4_-P. In the case of NO_3_-N, both upstream and downstream models when fitted with one predictor gave an R^2^ of 0.865 and 0.927, respectively, and when trained with all the predictors, both models resulted with an R^2^ of 0.999. There is a wide difference between the two models of OPO_4_-P when fitted with just one predictor, where the upstream model had an R^2^ of 0.342 and the downstream one had the value of 0.930. Nevertheless, the models performed comparably when fitted with all the predictors, with the R^2^ values of 0.989 and 0.959. The solo NH_4_-N model performed at R^2^ of 0.702 with one predictor and at R^2^ of 0.970 with all the predictors.

The models in general did not have a considerable increase in R^2^ after the addition of 5 predictors (Table 3). In some cases, with even fewer predictors there was no substantial improvement in R^2^, for example with NO_3_-N at both stations after 3 predictors. There are some predictors that repeat themselves in most cases. For example, week_sin is the first predictor in all the cases. EC is the second predictor in all the cases except 1. Similarly, Temp is the third predictor for all the cases except 1. DO is at the fourth or fifth position in 4 of the cases. Q is at the fourth or fifth position in 2 cases. Out of the 25 positions (first 5 predictors × 5 cases) 23 are taken by the same set of predictors i.e., week_sin, EC, Temp, DO and Q.

### 3.2. Comparison of Random Forest Model Performance with Linear Regression Model

The MLR and RF models fitted with their respective 5 best predictors were tested on the earlier separated unseen data. The 5 best predictors for the MLR model were determined in the same way as for the RF models. The results are reported in RMSE and R^2^ in Figure 1. The RF model performances on unseen data were typically comparable to that of the working datasets (comparing Table 3 and Figure 1). The RF models consistently outperformed the MLR models. The R^2^ values of the RF models were found to be 0.999 and 0.999 for NO_3_-N, 0.987 and 0.963 for OPO_4_-P and 0.983 for NH_4_-N, whereas for the MLR model they were found to be 0.886 and 0.909 for NO_3_-N, 0.752 and 0.59 for OPO_4_-P and 0.195 for NH_4_-N. For the Kahl am Main Station, the RF model reduced the RMSE in comparison to the MLR models for NO_3_-N from 0.266 to 0.024 mgN L^−1^, for OPO_4_-P from 0.021 to 0.005 mgP L^−1^, for NH_4_-N from 0.017 and 0.003 mgN L^−1^. Similarly, for the Erlabrunn Station, the RMSE values reduced for NO_3_-N from 0.296 to 0.025 mgN L^−1^ and for OPO_4_-P from 0.019 to 0.006 mgP L^−1^. These results suggest that RF models are capable of reproducing data trends and seasonality as well as providing reasonably accurate estimates of NO_3_-N, NH_4_-N, and OPO_4_-P concentrations.

### 3.3. Model’s Performance with Varying Sampling Frequencies

The RF models were run on the working dataset with 40 various sampling frequencies ranging from once a month to every 15 min. The relation of model performance as a function of sampling frequency is plotted in Figure 2. As the sampling frequency increases, the RMSE value goes down. The knee point in this case will be the sampling frequency after which any increase in it will not result in a significant increase in RMSE. The OSFs were found to be every 3.6 and 2.8 h for NO_3_-N at Kahl am Main and Erlabrunn stations, respectively. For OPO_4_-P, the OSFs were found to be every 2.4 and 1.8 h for the Kahl am Main and Erlabrun stations, respectively. For NH_4_-N, the OSF was every 2.2 h for the Kahl am Main station.

Figure 3, Figure 4 and Figure 5 show the time series comparing observed and predicted data when models were tested over the testing dataset after being trained on four different sampling frequencies: monthly, daily, optimal and 15-min. As the sampling frequency gets higher the predicted curve increasingly overlaps the observed curve. For monthly sampling frequency in all the three cases, the predicted curves mostly stay between the highs and lows of the observed curve. The models do a good job of predicting the seasonal patterns but fail to predict the peaks and troughs. As the sampling frequency gets higher, the peaks and troughs are reproduced increasingly better. For both stations, the predicted curves of NO_3_-N almost completely overlap the observed curves for the optimal and 15-min sampling frequencies (Figure 3c,d).

For OPO_4_-P and NH_4_-N, where the observed values are smaller than 0.2 mgP L^−1^ and 0.15 mgN L^−1^, respectively, the models need to be comparatively more sensitive than the NO_3_-N model to detect the decimal level changes. Their performances are shown in Figure 4 and Figure 5, respectively. By visual comparison of the time series at the OSF and 15-min interval, the shape of the plots looks very similar and they do not seem to differ much. Nevertheless, there are some peaks and troughs that are better predicted by the 15-min model. For example, in the NH_4_-N case (Figure 5), when comparing the OSF plot against the 15-min plot there is a trend of observed values peeking out at the peaks and bottoms. This peeking is much more prominent between January 2021 and July 2021, whereas for the 15-min graph there is much more overlap between the observed and predicted.

## 4. Discussion

### 4.1. Nutrient Soft Sensor Performance Using RF

The utilization of RF based soft sensors has demonstrated impressive predictive power, as evidenced by R^2^ values exceeding 0.95 in the prediction of NO_3_-N, OPO_4_-P and NH_4_-N. Our results are comparable to the similar study using 15-min high-frequency measurements by Yen et al. (2022) and have higher R^2^ values than the three studies mentioned in Section 1 [13,14,15]. Remarkably, this predictive capacity was maintained even with a smaller subset of predictors, comprising of five variables. These findings are consistent with the prior research [17,37] in this field, which likewise emphasized the importance of optimizing soft-sensor performance through the identification of optimal predictor sets. In this study it can be seen that the order of the inclusion is random and does not follow the order of the highest correlated variable going first as was previously expected (Appendix A). The RF model uses predictor variables that, a priori, are irrelevant, neither intuitively nor when looking at correlation coefficients or stepwise subsets with linear models. This is due to the intricate linkages that underlie the model and demonstrates the ability of the RF algorithm to mimic these non-linear relations embedded in the environmental phenomenon. The current work’s findings indicate that in order to successfully complete the variable sub-setting process, it is essential to follow a procedure that is in line with the model that will be implemented. In the literature there exist simpler and less resource-intensive methods such as feature weight coefficient or feature importance [38,39]. However, when the goal is to choose the fewest number of predictors without compromising the model’s accuracy, user-defined performance metrics perform better.

### 4.2. Optimal Predictor Subset and Sampling Freuqnecy

The best subgroups of the five predictors for all the soft sensors in both the stations only differ in one predictor: week_sin, EC, and Temp are there in all the five cases; DO is employed in two of the cases and Q in the other three. This indicates that a group of four predictors would be able to predict all the nutrients simultaneously with R^2^ greater than 0.99 for two cases and greater than 0.95 for the other three. EC, and Temp, DO and Q are commonly measured in rivers and streams by physical sensors within the authority monitoring schemes with high robustness, accuracy, and temporal resolution. Using them as surrogates would increase the applicability of setting up the soft sensors for nutrients in a broader context in the real world.

Another main finding of this study is that the performances of the nutrient surrogate models depend on the frequency of the data they are trained with; the higher the frequency, the better the performance in all the soft sensor cases. The graphs presenting the model performances with different sampling frequencies, as Figure 2 in this study, provide stakeholders with evidence to help them determine the level of accuracy and the corresponding sampling frequency required for a specific case. For example, when being trained on a monthly sampling frequency, commonly performed in routine monitoring schemes, the model for NO_3_-N at the Kahl am Main station gave an RMSE of 0.35 mgN L^−1^. In comparison, the RMSE was 0.02 mgN L^−1^ when the model was trained on the 15-min-interval data. Nevertheless, the model trained with the monthly data and their results can still be useful in certain cases. Overall, this study demonstrates the possible RMSEs when the surrogate models for nutrient soft sensors are trained on different sampling frequencies, offering a first idea of the accuracy level such models might achieve.

Furthermore, the results of this study show that the RF models can perform without losing much of their accuracy when trained at a lower frequency than 15-min intervals. In the cases of OPO_4_-P and NH_4_-N, the OSFs were found to be approximately every 2 h. By using these OSF, the R^2^ values when compared to those of the models trained on 15-min interval had a worsening of 2.75% and 1.69% in the case of OPO_4_-P at the upstream and downstream stations, respectively, and had a worsening of 3.72% for NH_4_-N at the downstream station. However, at those OSFs, the amount of data collected and stored is only 1/8 of what was previously being monitored at 15-min intervals. In the case of the NO_3_-N soft sensor at the Kahl am Main station, the OSF was found to be approximately every 4 h. By using this frequency, the model had a worsening of 0.25% in the R^2^ values compared to the 15-min ones. Similarly, for the Erlabrunn station, the OSF for NO_3_-N was determined to be approximately every 3 h and resulted in a compromise of 0.15% in the R^2^. These results would allow the physical sensors installed on the rivers to operate at lower frequencies and would save significant storage space and costs associated with measuring, handling and processing data for building soft sensors.

Coraggio et al. (2022) used the power spectral density (PSD) function to determine the frequencies with the highest density in the water quality measurements with high-frequency sensors [40]. This helped them determine how representative a particular frequency is of the overall situation. Their PSD cumulation curve showed that after a certain point, increasing the frequency does not provide any additional information, enabling them to identify a knee point for OSF, similar to the current study. They analyzed electrical conductivity, dissolved organic matter, dissolved oxygen, turbidity, and temperature measurements, and found their OSFs of 6, 5, 5, 3, and 9 h, respectively. These results are comparable with the present study, indicating that the OSF of the water quality data for training soft sensors is also nearly optimal for monitoring them. In other words, if a physical sensor is set up to accurately measure and provide a comprehensive understanding of what it is measuring, it will also provide an optimal amount of data for building soft sensors.

### 4.3. Measures against Overfitting

A big concern with random forest models is that they are highly susceptible to overfitting. The results of this study, with the R^2^ values higher than 0.95 in all cases, suggest the possibility of this issue. To make sure the models were not overfitted, in the first place, all models regardless of at which frequencies and which subsets of predictors were tested with 20% of the of the datapoints from the whole dataset. Furthermore, in some cases, the models using lower frequencies were tested on a much larger dataset than they were trained on. For example, the models, as shown in Figure 3b, Figure 4b and Figure 5b, were tested on a dataset that was 19 times larger than the training dataset, and they achieved the R^2^ values of 0.98, 0.72, and 0.73, respectively, and these R^2^ values point towards a low likelihood of overfitting. Additionally, visual inspection of the aforementioned figures also suggests against overfitting as the models perform well especially when considering the substantial difference between the size of the training and testing dataset. The models in the figures simulate the periodicity of the time series rather well, although they miss out on predicting the highs and lows.

### 4.4. Limination and Future Research Perspective

Although the models for NH_4_-N and OPO_4_-P produced R^2^ values of 0.95 and above, there should be a word of caution. Both sensors have a least count of 0.01 mg L^−1^, which means there is an uncertainty of 0.01 mg L^−1^ with each measurement. The mean concentrations of NH_4_-N and OPO_4_-P at the Kahl am Main station were 0.03 mgN L^−1^ and 0.10 mgP L^−1^, respectively, which inherently introduces an uncertainty of 33% and 10%. This high level of uncertainty makes the measurements from these sensors imprecise. For comparison, the mean of NO_3_-N is 3.66 mgN L^−1^ at the Kahl am Main station and is also measured with a sensor with a least count of 0.01 mgN L^−1^, which amounts to only 0.26% uncertainty. Since the measurements for NH_4_-N and OPO_4_-P are not that precise, the models might have also incorporated these patterns of errors. This problem is evident from the time series in Figure 4 and Figure 5, which are patchy with sudden peaks when compared to the relatively continuous time series of NO_3_-N (Figure 3). Therefore, for the development of reliable soft sensors for NH_4_-N and OPO_4_-P, there is a need for the development of physical sensors with higher precision for them.

The findings of this study would give the stakeholders the optimal number of physical sensors and their OSFs for the development of a soft sensor for measuring continuous real-time nutrients. However, it has not been verified whether the results are applicable for the long run or for large-scale monitoring networks. With a better understanding of the marginal return on sampling efforts, physical sensors and their sampling frequency can be changed to balance effort and collect more meaningful information. Following up on these findings, additional studies are suggested, including: (1) investigating the spatial variation of sampling sites and how it affects optimal sampling frequency, (2) investigating the spatial variation of sampling sites and how it affects optimal subset of predictors, and (3) building a generalized model for soft sensors which go beyond site-specific to larger and wider monitoring networks. This could be accomplished by following the idea of the present study i.e., applying the methodology from this study to more stations and comparing the results of optimal predictors and sampling frequencies. This approach is equally applicable to building soft sensors for other water quality parameters, such as turbidity, total dissolved solids, heavy metals and so on.

## 5. Conclusions

RF models demonstrate a significant capacity to build soft sensors for surrogate nutrient concentrations and support continuous high-frequency water quality monitoring. The RF models performed with R^2^ values higher than 0.95 for all the 5 cases, namely NO_3_-N, NH_4_-N, OPO_4_-P concentrations at 15-min intervals at 2 stations on the Main River, Germany. Feature engineering results demonstrated that the model could perform at good accuracy even when trained on fewer predictors. Specifically, with 5 out of 9 predictors the model performed with an R^2^ of higher than 0.99 for 3 cases and higher than 0.95 for the other 2 cases. This testifies to the hypothesis of using a smaller subset of predictors to harnesses the complex relationships behind these models in a more efficient way. The best trade-off between performance and measurement frequency was found for the RF models utilizing knee point analysis since high measurement frequency raises issues with data handling and storage. The OSFs were found to be approximately 3 h and 4 h for NO_3_-N for the upstream and downstream stations, respectively, 2 h for OPO_4_-P for both the stations and 2 h for NH_4_-N for the upstream station. Overall, the findings of this study demonstrate that the model can perform well even when trained on a lower sampling frequency and with fewer predictors. This would ensure an overall leaner soft sensor for monitoring water quality in river systems with fewer storage problems and reduced operational costs.

## Figures and Tables

**Figure 1 sensors-23-06057-f001:**
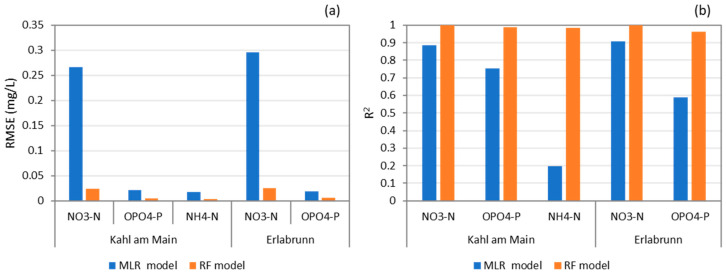
Performance of linear regression and random forest model on unseen data with (**a**) RMSE and (**b**) R^2^ as the evaluation criteria.

**Figure 2 sensors-23-06057-f002:**
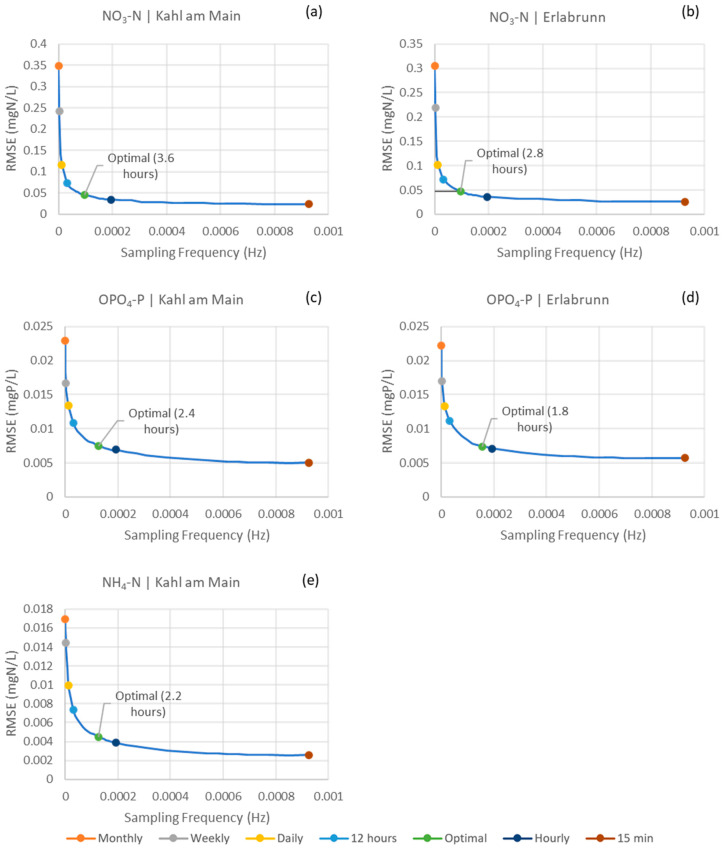
RMSE of the RF models as a function of sampling frequency in Hz: (**a**) NO_3_-N in Kahl am Main station; (**b**) NO_3_-N in Erlabrunn station; (**c**) OPO_4_-P in Kahl am Main station; (**d**) OPO_4_-P in Erlabrunn station; (**e**) I NH_4_-N in Kahl am Main station.

**Figure 3 sensors-23-06057-f003:**
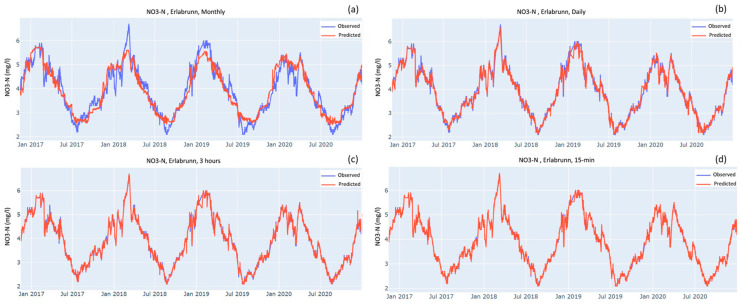
Observed and predicted values of NO_3_-N at Erlabrunn station when model was trained on the following frequencies: (**a**) monthly; (**b**) daily; (**c**) Optimal i.e., 2.8 h but, for the sake of simplicity, rounded off to 3 h; (**d**) 15 min.

**Figure 4 sensors-23-06057-f004:**
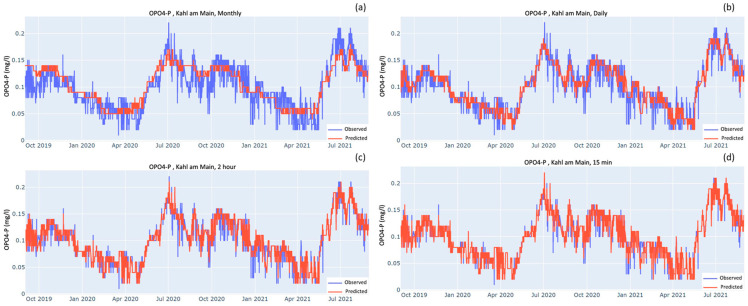
Observed and predicted values of OPO_4_-P at Kahl am Main station when the model was trained on the following frequencies: (**a**) monthly; (**b**) daily; (**c**) Optimal i.e., 2.4 h but, for the sake of simplicity, rounded off to 2 h; (**d**) 15-min intervals.

**Figure 5 sensors-23-06057-f005:**
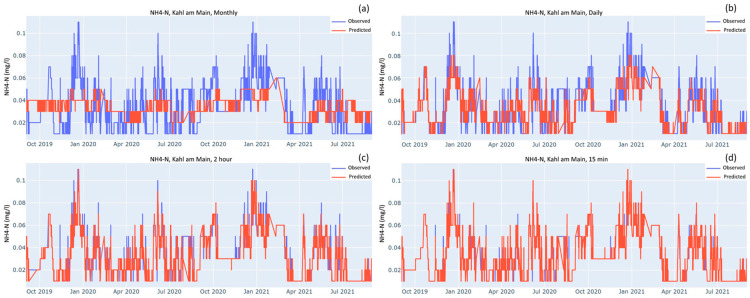
Observed and predicted values of NH_4_-N at Kahl am Main station when model was trained on the following frequencies: (**a**) monthly; (**b**) daily; (**c**) Optimal i.e., 2.2 h but for the sake of simplicity, rounded off to 2 h; (**d**) 15-min intervals.

**Table 1 sensors-23-06057-t001:** Mean, standard deviation (std), minimum value (Min) and maximum value (Max) of both the stations.

Parameter	Unit	Kahl am Main			Erlabrunn		
		Mean ± Std	Min	Max	Mean ± Std	Min	Max
DO	mg/L	9.63 ± 2.56	4.20	17.40	10.6 ± 2.13	5.30	16.30
Temp	°C	14.56 ± 7.14	1.10	27.60	13.49 ± 7.49	0.10	28.60
pH		8.02 ± 0.19	7.50	8.50	7.94 ± 0.20	7.40	8.50
Flow	m^3^/s	136.52 ± 73.95	100.00	518.00	145.98 ± 42.80	39.80	437.0
EC	µS/cm	616.95 ± 69.87	405.00	751.00	639.42 ± 80.68	387.00	842.00
NO_3_-N	mgN L^−1^	3.66 ± 0.79	2.10	5.50	3.87 ± 0.98	2.09	6.70
OPO_4_-P	mgP L^−1^	0.10 ± 0.04	0.01	0.22	0.09 ± 0.03	0.02	0.16
NH_4_-N *	mgN L^−1^	0.03 ± 0.02	0.01	0.11			

* NH_4_-N is not measured at Erlabrunn station.

**Table 2 sensors-23-06057-t002:** Values of hyperparameters used in GridSearchCV.

Hyperparameter	Values
Bootstrap	True, False
Size of the random subsets	Auto, ‘sqrt’
Depth of the trees	10, 20, 30
Minimum number of samples to split a node	6, 12, 20
Minimum number of samples to be at a leaf node	6, 12, 20

**Table 3 sensors-23-06057-t003:** Cross-validation results for the RF model with varying subset of predictors.

Order of Variables	1st	2nd	3rd	4th	5th	6th	7th	8th	9th
**Station: Kahl am Main (downstream)**									
**NO_3_-N**	+week_sin	+EC	+Temp	+DO	+month_sin	+pH	+week_cos	+month_cos	+Flow
R^2^	0.927	0.979	0.995	0.998	0.998	0.999	0.999	0.999	0.999
RMSE	0.676	0.587	0.397	0.205	0.097	0.043	0.034	0.028	0.026
**OPO_4_-P**	+week_sin	+EC	+Temp	+Flow	+DO	+month_cos	+week_cos	+month_sin	+pH
R^2^	0.930	0.973	0.985	0.988	0.990	0.990	0.989	0.989	0.989
RMSE	0.054	0.047	0.025	0.016	0.016	0.008	0.007	0.006	0.006
**NH_4_-N**	+week_sin	+EC	+Temp	+DO	+Flow	+pH	+month_sin	+month_cos	+week_cos
R^2^	0.702	0.907	0.957	0.968	0.970	0.970	0.971	0.971	0.970
RMSE	0.023	0.021	0.012	0.011	0.007	0.006	0.006	0.005	0.004
**Station: Erlabrunn (upstream)**									
**NO_3_-N**	+week_sin	+EC	+Temp	+Flow	+pH	+month_cos	+week_cos	+month_sin	+DO
R^2^	0.865	0.968	0.994	0.998	0.999	0.999	0.999	0.999	0.999
RMSE	1.001	0.837	0.526	0.162	0.097	0.063	0.063	0.043	0.037
**OPO_4_-P**	+week_sin	+Flow	+EC	+Temp	+DO	+week_cos	+pH	+month_cos	+month_sin
R^2^	0.342	0.848	0.929	0.954	0.956	0.958	0.959	0.959	0.959
RMSE	0.035	0.031	0.022	0.019	0.013	0.013	0.010	0.008	0.007

## Data Availability

The dataset used in this study are publicly available and be accessed from https://www.gkd.bayern.de.

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
