# Peer review of "Determination of Optimal Predictors and Sampling Frequency to Develop Nutrient Soft Sensors Using Random Forest"

_sensors, 2023, doi:10.3390/s23136057_

Round 1

Reviewer 1 Report

Dear Autor:

this study explores the use of soft sensors based on data-driven models as an alternative to direct nutrient measurements, which are often challenging and expensive. By identifying the optimal subset of predictors and the sampling frequency for developing nutrient soft sensors using random forest, you have demonstrated the potential for these models to support continuous high-frequency water quality monitoring. 

This research involved using water quality data from two automatic stations on the Main River in Germany and included dissolved oxygen, temperature, conductivity, pH, streamflow, and cyclical time features as predictors. By using forward subset selection, you identified the optimal subset of predictors and found that the models fitted with these predictors produced R2 values above 0.95 for nitrate, orthophosphate, and ammonium for both stations. 

They then trained the models on 40 different sampling frequencies, ranging from monthly to 15-minute intervals, and found that the model's performance improved as the sampling frequency increased. By using a knee-point determination algorithm, you identified the optimal balance between sampling frequency and model performance, which was found to be approximately 3 hours and 4 hours for NO3-N for the upstream and downstream stations respectively, 2 hours for OPO4-P for both the stations, and 2 hours for NH4-N for the upstream station.

these findings suggest that soft sensors for monitoring nutrient levels can function with fewer predictors at lower frequencies without a significant decrease in performance. This has important implications for reducing operational costs and storage problems associated with high-frequency data handling. 

Overall, your study demonstrates the significant potential of RF models in building soft sensors for surrogate nutrient concentrations. Your findings have important implications for water quality monitoring in river systems and highlight the potential for more efficient and cost-effective nutrient monitoring using soft sensor technology. 

Suggestions for research are presented: 

Based on the findings of the study, further research could be conducted to explore the use of soft sensors for monitoring nutrient levels in other water bodies and under different environmental conditions. The study focused on water quality data from two automatic stations on the Main River in Germany, and it would be interesting to see if similar results can be achieved in other rivers or lakes.

Additionally, the study used forward subset selection to identify the optimal subset of predictors for developing nutrient soft sensors using random forest. Other feature selection techniques could be explored to compare the performance of the models and identify the most effective method for developing soft sensors for monitoring nutrient levels.

Moreover, the study used a knee-point determination algorithm to identify the optimal balance between sampling frequency and model performance. Other methods could be explored to determine the optimal sampling frequency, such as machine learning algorithms or statistical methods.

Finally, the study focused on developing soft sensors for monitoring nutrient levels. Future research could explore the use of soft sensors for monitoring other water quality parameters, such as turbidity, total dissolved solids, or heavy metals. This could provide a more comprehensive picture of water quality and improve our ability to monitor and manage water resources.

Author Response

Response to Reviewer 1

Suggestions for research are presented:

  1. Based on the findings of the study, further research could be conducted to explore the use of soft sensors for monitoring nutrient levels in other water bodies and under different environmental conditions. The study focused on water quality data from two automatic stations on the Main River in Germany, and it would be interesting to see if similar results can be achieved in other rivers or lakes.

Response:

We agree with the reviewer’s comment in general. One of the main aims of this publications is to promote similar studies at different locations. We have suggested three future research directions following this study and the first one resembles this very suggestion of the reviewer; see line 447.

  1. Additionally, the study used forward subset selection to identify the optimal subset of predictors for developing nutrient soft sensors using random forest. Other feature selection techniques could be explored to compare the performance of the models and identify the most effective method for developing soft sensors for monitoring nutrient levels.

Response:

The reason for using the forward subset selection to identify the optimal subset of predictors is its interpretability. The method for the subset selection wouldn’t overall affect the performance of the model. One study to mention here which has a very similar research objective and uses a different method for the predictor subset selection is Castrillo et al. (2020), they used backward subset selection and can be referred for further details.  

  1. Moreover, the study used a knee-point determination algorithm to identify the optimal balance between sampling frequency and model performance. Other methods could be explored to determine the optimal sampling frequency, such as machine learning algorithms or statistical methods.

Response:

Thank you for the comment. Yes, in general there are other methods which can be used for determining the optimal sampling frequency, which we mentioned in Line 99 – 112 in the manuscript. After careful literature review, we have decided to use the knee-point determination algorithm for this study as this method has been successfully applied for determining the OSF for the case of water quality parameters in the most recent study in this area done by Coraggio et al. (2022).

  1. Finally, the study focused on developing soft sensors for monitoring nutrient levels. Future research could explore the use of soft sensors for monitoring other water quality parameters, such as turbidity, total dissolved solids, or heavy metals. This could provide a more comprehensive picture of water quality and improve our ability to monitor and manage water resources.

Response:

Thank you for the suggestion. We fully agree with the reviewer’s comment. We have added one sentence explicitly addressing this point; see the lines 466-468 in the revised manuscript.

Reference

Castrillo, M. and Á.L. García, Estimation of high frequency nutrient concentrations from water quality surrogates using machine learning methods. Water Research, 2020. 172: p. 115490

Coraggio, E., et al., Water Quality Sampling Frequency Analysis of Surface Freshwater: A Case Study on Bristol Floating Harbour. Frontiers in Sustainable Cities, 2022. 3.

Reviewer 2 Report

The topic of the paper and the result are interesting. However, some points must be clarified, and the paper writing must be significantly modified in order to be accepted as a publication. Please kindly consider the following suggestions.

  1. Please explain the structure of the paper in the last paragraph of section 1 introduction.
  2. In introduction, the authors wrote that some studies (ref [12]-17])already proposed the utilization of random forest (RF) model in this field. Please explain in introduction the difference  in of your proposed method  approach compared to the previous studies ([12]-[17])
  3. There are no explanation about the predictors considered in this research, such as week_sin, EC, Temp, DO and Q. Please summarize the meaning of each predictor in table.
  4. I suggest to remove the graphic title for Figure 3-5 because it is very small and redundant with the explanation in figure caption.
  5. For some cases, it is difficult to compare the performance of prediction among the simulated frequencies in figure 3-5. For example, the figure 3 c and d almost look like the same. Other than visual look as shown by Fig 3-5, I suggest the authors to summarize the performance of prediction in figure 3-5 using root mean squared error (RMSE).
  6. As introduced in section 1, some studies already proposed the utilization of RF model in this field. To insist the significance of this study’s result, It’s better if the authors can show the prediction performance comparison between their propose RF model with other papers RF model. Currently, the authors only compared it with multi linear regression 

Author Response

Response to Reviewer 2

The topic of the paper and the result are interesting. However, some points must be clarified, and the paper writing must be significantly modified in order to be accepted as a publication. Please kindly consider the following suggestions.

  1. Please explain the structure of the paper in the last paragraph of section 1 introduction.

Response:

Thank you for the comment. The last paragraph of Section 1 introduces the aim, the specific objectives, and the significance of the study following the common writing protocols. Although the structure the paper is not explicitly mentioned, we have tried to guide the readers through well marked subheadings. In the revised version we’ve also added subheadings in the discussion section, giving more structure to the manuscript.

  1. In introduction, the authors wrote that some studies (ref [12]-17])already proposed the utilization of random forest (RF) model in this field. Please explain in introduction the difference  in your proposed method  compared to the previous studies ([12]-[17])

Response:

Thanks for the comment. The studies mentioned ([12] - [17]) establish the performance of the RF model for cases like these. Based on the good performance of RF models for water quality cases, this study focuses on making the RF model even more efficient. The study does it by quantifying the exact amount of data needed for the models proper training. We don’t aim to build the random forest in a different way but rather build up on the hypothesis already established by these studies that RF performs good and work towards reducing the dependence of RF on extensive amount of data

  1. There are no explanation about the predictors considered in this research, such as week_sin, EC, Temp, DO and Q. Please summarize the meaning of each predictor in table.

Response:

The full names of EC, Temp, DO and Q are mentioned in the lines 130 – 132 upon their first appearance in the manuscript. They are commonly used in the water quality field. The explanation of the week_sin is in the lines 184-187 in the manuscript. To explicitly mention those four time predictors, we added one sentence in the lines 191-192  in the revised manuscript.

  1. I suggest to remove the graphic title for Figure 3-5 because it is very small and redundant with the explanation in figure caption.

Response:

Thank you for the suggestion. We would prefer keeping the graphic titles because it is ease of readability and findability in the figures.

  1. For some cases, it is difficult to compare the performance of prediction among the simulated frequencies in figure 3-5. For example, the figure 3 c and d almost look like the same. Other than visual look as shown by Fig 3-5, I suggest the authors to summarize the performance of prediction in figure 3-5 using root mean squared error (RMSE).

Response:

Thank you for the suggestion. The RMSE values of the RF models with inputs at different sampling frequencies are provided in Figure 2 and in the manuscript. Also, in reviewer 5’s comment we’ve already been made aware of the length of the paper. But for your reference we’ve added the table in the supplementary materials

  1. As introduced in section 1, some studies already proposed the utilization of RF model in this field. To insist the significance of this study’s result, It’s better if the authors can show the prediction performance comparison between their propose RF model with other papers RF model. Currently, the authors only compared it with multi linear regression 

Response:

Thank you for this suggestion. We have mentioned the performances of the RF models in this field in the introduction section. For a better comparison, we provided an addition in the lines 353-355 in the discussion section in the revised manuscript.

Reviewer 3 Report

The authors use machine learning to predict nutrient concentrations in stream water from other continuous measurements.  They focus on choosing the optimal, minimal number of predictors and identifying the optimal sampling frequency (OSF).  To identify OSF, the authors trained the model on subsamples of the data at different intervals, then tested with the 15 minute data (but see questions below).  The study provides a case study of the application of random forests to estimate nutrient concentrations from other water quality variables.  The innovations are modest but useful, and as a template of a specific workflow, including software used, it could be useful to others seeking to apply similar methods.

The writing is good overall, but often overly wordy.  Cut words.  I have provided some examples, but leave it to the authors to find other opportunities to cut and improve the writing.

L232 and elsewhere:  “surrogating” is not the correct term.  Maybe “estimating”.

L232-236.  Please clarify how the datasets at different sampling frequencies were generated.  See pdf for more detail.

L232-235.  So the model was trained with coarser resolution date (e.g monthly), then validated on the same 15-minute data used to evaluate the original 15-minute model? 

Figure 3.  To me, these are the most useful plots and could come before Figure 2.  Please also include a hydrograph of discharge, so we can see how flow varied during the period.  Does flow vary gradually at this site, or are there peak flows that were successfully modelled?

See additional comments, questions and suggestions in the annotated PDF.

See comments above.

Author Response

Response to Reviewer 3

The authors use machine learning to predict nutrient concentrations in stream water from other continuous measurements.  They focus on choosing the optimal, minimal number of predictors and identifying the optimal sampling frequency (OSF).  To identify OSF, the authors trained the model on subsamples of the data at different intervals, then tested with the 15 minute data (but see questions below).  The study provides a case study of the application of random forests to estimate nutrient concentrations from other water quality variables.  The innovations are modest but useful, and as a template of a specific workflow, including software used, it could be useful to others seeking to apply similar methods.

The writing is good overall, but often overly wordy.  Cut words.  I have provided some examples, but leave it to the authors to find other opportunities to cut and improve the writing.

Response:

Thank you for the editing. We have gone through the whole revised version of the reviewer, and made the changes in the revised manuscript. These changes have been highlighted to facilitate easier viewing.

  1. L232 and elsewhere:  “surrogating” is not the correct term.  Maybe “estimating”.

Response:

Thank you for the comment. We have corrected here to estimating.

  1. L232-236.  Please clarify how the datasets at different sampling frequencies were generated.  See pdf for more detail.

Response:

Thank you for the comment. For more clarity, we have elaborated the sampling methods with an example of how we sampled the dataset with hourly interval from the 15-min interval in the lines 236-239 in the revised version.  

  1. L232-235.  So the model was trained with coarser resolution date (e.g monthly), then validated on the same 15-minute data used to evaluate the original 15-minute model? 

Response:
Yes, the model was trained on coarser resolution and tested on the same dataset used to test the 15-minutes model.

  1. Figure 3.  To me, these are the most useful plots and could come before Figure 2.  Please also include a hydrograph of discharge, so we can see how flow varied during the period.  Does flow vary gradually at this site, or are there peak flows that were successfully modelled?

Response:

Thank you for the comment. We acknowledge the significance of including a hydrograph in a study like this. However, in our research, all the other predictors hold equal, if not greater, importance than the hydrograph. Due to space constraints, it is not feasible to include all the graphs in the manuscript. If the readers are interested in the time series of the variables, they can find them from the Waterscience service Bavaria (Gewässerkundlicher Dienst Bayern, https://www.gkd.bayern.de/). The data source is explicitly mentioned in the manuscript. By using the optimal sampling frequency and 15-min resolution dataset, the model performance does not differ between high and low flows (Figs. 3-5).

Reviewer 4 Report

Comments to Authors:

In this study, the authors developed a robust and efficient soft sensor for the surrogating high-frequency nutrient concentrations using other physical sensor measurements. Overall, the study is methodologically sound with promising results. However, prior to further consideration, some comments to be addressed:

(i) Introduction: Highlight the research gaps. Research objectives have been written but still the gap is not too clear. Perhaps you can incorporate following literature to enrich the introduction:

For IoT: Toward industrial revolution 4.0: Development, validation, and application of 3D-printed IoT-based water quality monitoring system

For overall water quality: Comparison among different ASEAN water quality indices for the assessment of the spatial variation of surface water quality in the Selangor river basin, Malaysia

(ii) Methodology: Provide a flowchart of your research. It is not easy to understand the flow at the moment.

(iii) Results section: Interesting however seems to be confusing. If I do not follow the sequence of addition of parameters: For instance, for No3-N, I add EC, then flow, then week_cos, I believe the results will be different. Therefore, you might need to provide some statistical reasons for using this method to select predictors.

Discussion section: Divide them into sections. Too lengthy and hard to identify the focus point of the discussion. For instance: Applicability of RF-based soft sensors for first two paragraphs. Also, incorporate your discussion with other studies. 
For instance: (iii) Application of artificial intelligence methods for monsoonal river classification in Selangor river basin, Malaysia discussed the reasons why NN outperformed SVM, which can be included in your studies.

Also, (iv) Water quality index modeling using random forest and improved SMO algorithm for support vector machine in Saf-Saf river basin which can be cited for better comparison.

(v) Conclusion: Conclusion section seems to be a repetition of the results section. Huge modifications are required. Please provide insights into this study and what can be further done in the future.

(vi) Implications for future research may also be included in the conclusion at the end.

Proofreading is necessary.

Author Response

Response to Reviewer 4

In this study, the authors developed a robust and efficient soft sensor for the surrogating high-frequency nutrient concentrations using other physical sensor measurements. Overall, the study is methodologically sound with promising results. However, prior to further consideration, some comments to be addressed:

(i) Introduction: Highlight the research gaps. Research objectives have been written but still the gap is not too clear. Perhaps you can incorporate following literature to enrich the introduction:

For IoT: Toward industrial revolution 4.0: Development, validation, and application of 3D-printed IoT-based water quality monitoring system

For overall water quality: Comparison among different ASEAN water quality indices for the assessment of the spatial variation of surface water quality in the Selangor river basin, Malaysia

Response:

Thank you for the comment. The objectives are mentioned in the last paragraph of the introduction section. The research gap is explicitly mentioned in Line 112 to 113, emphasizing the knowledge gap of optimal sampling frequency for building water quality soft sensors. We find the first literature the reviewer suggested is helpful to enrich the introduction, therefore it is cited in the line 38 in the revised manuscript.

(ii) Methodology: Provide a flowchart of your research. It is not easy to understand the flow at the moment.

Response:

Thank you for the suggestion. We have supplemented the workflow chart for more clarity in Fig. S1 of supporting information and one sentence referring to the flowchart is supplemented in the line 134 in the revised version.

(iii) Results section: Interesting however seems to be confusing. If I do not follow the sequence of addition of parameters: For instance, for No3-N, I add EC, then flow, then week_cos, I believe the results will be different. Therefore, you might need to provide some statistical reasons for using this method to select predictors.

Response:

Thank you for the comment. The combination of NO3-N, EC, flow and week_cos will always give the same performance irrespective of the sequence of the input predictors. However, we did the sub-setting to figure out what’s the smallest, best combination of the predictors. The underlying statistical method that defines which predictor will be added next is the increment in R2 as mentioned in the line 259. The next predictor is defined after trying out all the possible combinations. The reason for using the forward subset selection to identify the optimal subset of predictors is its interpretability.

Discussion section: Divide them into sections. Too lengthy and hard to identify the focus point of the discussion. For instance: Applicability of RF-based soft sensors for first two paragraphs. Also, incorporate your discussion with other studies.

For instance: (iii) Application of artificial intelligence methods for monsoonal river classification in Selangor river basin, Malaysia discussed the reasons why NN outperformed SVM, which can be included in your studies.

Also, (iv) Water quality index modeling using random forest and improved SMO algorithm for support vector machine in Saf-Saf river basin which can be cited for better comparison.

Response:

Thank you for the nice suggestions. We have divided the discussion section into 4 subsections and corresponding titles; see in the revised version. We found that the literatures suggested by the reviewer are difficult to be incorporated into the discussion section.

(v) Conclusion: Conclusion section seems to be a repetition of the results section. Huge modifications are required. Please provide insights into this study and what can be further done in the future.

(vi) Implications for future research may also be included in the conclusion at the end.

Response:

Thank you for the suggestions. The conclusion section summarizes the main results of the manuscript, which offers the readers a recap of the take-home messages of this study. The implications for future research are mentioned explicitly in the last paragraph of the discussion. Also, one more scope for the future research is supplemented in the lines 466-468 in the revised version.

Reviewer 5 Report

Application of Artificial Neural Network (ANN) for investigation of the impact of past and future land use land cover change on streamflow in the Upper Gilgel Abay watershed, Ethiopia

Dear Authors

The basic science of this paper is conducted in a good way and is an appropriate standard.  The author and his team write this paper according to the journal's scope. I already published several papers in this domain. There are many flaws in this manuscript.  I am glad to review this paper because this manuscript is very relevant according to my research.  To address this, the study aimed to determine the optimal subset of predictors and the sampling frequency for developing nutrient soft sensors using random forest. The study used water quality data at 15-minute intervals from two automatic stations on the Main River, Germany, and included dissolved oxygen, temperature, conductivity, pH, streamflow, and cyclical time features as predictors. The optimal subset of predictors was identified using forward subset selection, and the models fitted with the optimal predictors produced R2 values above 0.95 for nitrate, orthophosphate, and ammonium for both stations.  At this stage, I am just suggesting a major revision of this paper because there is no novelty and the structure of this paper is not appropriate. 

Major comments

·         Title is very lengthy and it’s not appropriate.

·         Abstract is inappropriate and there is no significance according to international journal criteria.

·         Check the abstract as well as the whole manuscript. There are many syntax error problems. Remove

·         Moreover, the abstract section does not reflect the whole research.

·         Rewrite the whole abstract section because the abstract section does not reflect the whole study. Moreover, the abstract section is very complex and there is no continuity in the sentences. Why the author used this study, the author should focus on the main aim of the research

·         Introduction section is very lengthy and there is no sequence of this research.

·         Recheck the language of the whole manuscript.

·         Introduction section is not appropriate and the problem statement, research questions, and hypothesis are missing from the introduction section.

·         Introduction section is very simple and I did not find the research question, problem statement, and innovative idea of this research.

·         Why author used the figure caption inside the figures?

·         Modify all figures according to international journal criteria and use the same font style.

·         Check the typo and syntax error mistakes in the whole manuscript during revision.

·         Add study area figure with study area

·         Split data from the study area.

·         Add a flowchart in the methodology section.

·         Split results and discussion section

·         Modify the quality of figures

·         Add recommendations and future work in the conclusion section

Best Regards

Author Response

Response to Reviewer 5

Dear Authors

The basic science of this paper is conducted in a good way and is an appropriate standard.  The author and his team write this paper according to the journal's scope. I already published several papers in this domain. There are many flaws in this manuscript.  I am glad to review this paper because this manuscript is very relevant according to my research.  To address this, the study aimed to determine the optimal subset of predictors and the sampling frequency for developing nutrient soft sensors using random forest. The study used water quality data at 15-minute intervals from two automatic stations on the Main River, Germany, and included dissolved oxygen, temperature, conductivity, pH, streamflow, and cyclical time features as predictors. The optimal subset of predictors was identified using forward subset selection, and the models fitted with the optimal predictors produced R2 values above 0.95 for nitrate, orthophosphate, and ammonium for both stations.  At this stage, I am just suggesting a major revision of this paper because there is no novelty and the structure of this paper is not appropriate. 

Major comments

  • Title is very lengthy and it’s not appropriate.

Response:

Thank you for the comment. We believe to represent the whole study through the title the length is a result of that.

  • Abstract is inappropriate and there is no significance according to international journal criteria.

Response: Thank you for the input. We believe an abstract consists of motivation for the study, concise summary of the methodology, important results and the conclusion in the broader context of the researchwe . We have tried to include all those elements.

  • Check the abstract as well as the whole manuscript. There are many syntax error problems. Remove

Response:

Thank you for the comment. We have proofread the manuscript and corrected some errors; see the abstract section in the revised manuscript.

  • Moreover, the abstract section does not reflect the whole research.

Response:

Our abstract contains the motivation for the study, gist of methodology, important results, and the conclusion in the broader context of the research and therefore we believe the abstract reflect the whole research contents as responded to another comment above.

  • Rewrite the whole abstract section because the abstract section does not reflect the whole study. Moreover, the abstract section is very complex and there is no continuity in the sentences. Why the author used this study, the author should focus on the main aim of the research

Response:

The contents of this comment have been addressed in the response above.

  • Introduction section is very lengthy and there is no sequence of this research.

Response: Thank you for the comment. The sequence that we have aimed for in the introduction is as follows:

Paragraph 1: The significance of monitoring nutrients and its importance.
Paragraph 2: Overview of the current sensors used and their limitations.
Paragraph 3: Introduction to soft sensors as a potential solution and highlighting previous studies that have developed high-performing soft sensors.
Paragraph 4: Explaining the benefits of defining a subset of predictors in enhancing the overall efficiency of soft sensors.
Paragraph 5: The necessity of determining the optimal sampling frequency for soft sensors, considering their current requirement for high-frequency data.
Paragraph 6: Methods used to determine the optimal sampling frequency for sensors, along with a mention of the existing research gap in this area.
Paragraph 7: Stating the aim of the study.

We believe that this sequence effectively provides the necessary background information and justifies the relevance and importance of the study.

  • Recheck the language of the whole manuscript.

Response:

Thanks for the comment. We have proofread the whole manuscript and made corrections accordingly.

  • Introduction section is not appropriate and the problem statement, research questions, and hypothesis are missing from the introduction section.

Response:

Thanks for the comment. The research objectives are mentioned in the last paragraph of the introduction section. The problem statement is explicitly mentioned in the lines 113 and 114, emphasizing the knowledge gap of optimal sampling frequency for building water quality soft sensors.

  • Introduction section is very simple and I did not find the research question, problem statement, and innovative idea of this research.

Response:

The contents of this comment have been addressed in the response above.

  • Why author used the figure caption inside the figures?

Response:

Thank you for the comment. We would prefer keeping the graphic titles because it is ease of readability and findability in the figures.

  • Modify all figures according to international journal criteria and use the same font style.

Response:

Thanks for the comment. From our perspective, the figures are of adequate quality. Despite the small size and abundance of information in Figs. 3-5 maintain clarity and do not pixelate even upon extensive zooming.

  • Check the typo and syntax error mistakes in the whole manuscript during revision.

Response:

Thank for the comment. We have proofread the whole manuscript and made corrections accordingly.

  • Add study area figure with study area

Response:

Thanks for the suggestion. A figure about the study area is added to SI Fig. S2 according to your suggestion. Also, the Fig. S2 is mentioned in the revised manuscript.

  • Split data from the study area.

Response:

Thank you for the suggestion. The datasets are needed to be introduced together with the stations because they are site-specific. Therefore, we have decided to combine them in one subsection.

  • Add a flowchart in the methodology section.

Response:

Thanks for the comment. We have supplemented the workflow chart for more clarity in Fig. S1 of supporting information and one sentence referring to the flowchart is supplemented in the line 134 in the revised version.

  • Split results and discussion section

Response:

Thank you for the comment. The result section was split into 3 sections in the original manuscript. According to your suggestion, we have divided the discussion section into 4 subsections and corresponding titles; see in the revised version.

  • Modify the quality of figures

Response:

Thanks for the comment. From our perspective, the figures are of adequate quality. Even the Figs 3-5 are very small and abundance of information, they maintain clarity and do not pixelate even upon extensive zooming.

  • Add recommendations and future work in the conclusion section

Response:

Thanks for the comment. The perspective for future research is mentioned explicitly in the last paragraph of the discussion. Also, one more future research option is supplemented in the lines 461-463 in the revised version.

Round 2

Reviewer 1 Report

The authors have reviewed and answered the comments well

Reviewer 4 Report

I have carefully gone through the revised manuscript and I feel the authors have substantially addressed them accordingly.

I would recommend a "Acceptance" to the revised manuscript.

Minor editing of English language required

Reviewer 5 Report

the author revised the paper according to my comments and suggestions.